# Synthesis of Bimetallic FeCu-MOF and Its Performance as Catalyst of Peroxymonosulfate for Degradation of Methylene Blue

**DOI:** 10.3390/ma15207252

**Published:** 2022-10-17

**Authors:** Huanxuan Li, Chen Xu, Ning Li, Tao Rao, Zhong Zhou, Qingwei Zhou, Chunhui Wang, Shaodan Xu, Junhong Tang

**Affiliations:** College Materials & Environmental Engineering, Hangzhou Dianzi University, Hangzhou 310018, China

**Keywords:** FeCu-MOF, peroxymonosulfate (PMS), methylene blue (MB), reaction kinetics, mechanisms

## Abstract

Bimetallic MOFs have recently emerged as promising materials for wastewater treatment based on advanced oxidation processes. Herein, a new bimetallic MOF (FeCu-MOF) was fabricated by hydrothermal process. The structural, morphological, compositional and physicochemical properties of the as-synthesized bimetallic FeCu-MOF were characterized by XRD, FT-IR, SEM, TEM, BET, and XPS. TEM and XPS confirmed the homogeneous distribution of CuO_2_ nanoparticles in the as-synthesized materials. The result of wastewater treatment indicated that 100% of MB was removed by 6.0 mM PMS activated with 0.6 g/L of FeCu-MOF in 30 min. The high catalytic performance of FeCu-MOF was probably due to the accelerated electron and mass transfer resulting from the existence of a homogeneous distribution of unsaturated metal sites and an abundant mesoporous structure. The obtained results from the competitive quenching tests demonstrated that sulfate radicals (SO_4_^•^^−^) were the major species responsible for MB oxidation. In addition, hydroxyl (**·**OH) and singlet oxygen (^1^O_2_) also had a nonnegligible role in the MB removal. Interestingly, the addition of acetate ion (CHCOO^−^) promoted the removal of MB while other anions (including NO_2_^−^, H_2_PO_4_^−^, SO_4_^2^^−^, HPO_4_^2^^−^, and HCO_3_^−^) inhibited the MB removal. Furthermore, a possible mechanism based on both heterogeneous and homogeneous activation of PMS was proposed, along with the MB degradation mechanism.

## 1. Introduction

The importance of clean water resources for the sustainable development of society is self-evident. Over the past few decades, rapid population growth and industrial development have exacerbated the water crisis [1]. Removing pollutants from wastewater is one of the most effective ways to access clean water [2]. Organic dyes, which are widely used in printing, papermaking, textile, plastics, leather tanning, cosmetics, and other industries, are among the most significant forms of industrial wastewater in China [3,4,5]. Although the presence of organic dyes is not highly toxic in the environment, they still cause many adverse effects on organisms, such as mutagenesis, carcinogenicity, respiratory toxicity, and teratogenicity [6,7,8]. Methylene blue (MB), is one of the most widely used water-soluble azo dyes in industrial production and results in the generation of high-color and poorly biodegradable wastewater, which is difficult to completely remove from the aqueous medium by conventional methods [9].

Advanced oxidation processes (AOPs) based on peroxymonosulfate (HSO_5_^-^,PMS) as effective ways for the removal of refractory organic pollutants have attracted increasing interest among researchers [10,11,12]. Compared to hydrogen peroxide (H_2_O_2_) and peroxydisulfate (S_2_O_8_^2−^, PDS), PMS is a cost-effective, high-efficiency, and environmentally friendly oxidant. According to previous literature, the removal of organic pollutants in PMS-involved systems mostly relies on the generation of reactive species, including sulfate radicals (SO_4_^•^^−^), hydroxyl radicals (**·**OH), and singlet oxygen (^1^O_2_) [9,13]. However, PMS is relatively stable at room temperature and pressure, and additional energy such as light (Equation (1)) and heat (Equation (2)) or a catalyst (Equation (3)) is needed to break the O-O bond in its molecular structure and produce SO_4_^•^^−^ and/or **·**OH. SO_4_^•^^−^ (2.5–3.1 V) and **·**OH (1.8–2.7 V) feature high redox potential under neutral conditions, which can degrade most of the refractory organic pollutants in water [14,15,16]. Due to the advantages of easy operation and simple equipment, transition-metal-based catalysts are considered popular ways to activate PMS for the removal of pollutants. Among them, the Co(II) ion exhibits the best catalytic activity towards PMS, but iron- and copper-based catalysts as activators of PMS have been widely studied due to their huge abundance and environmental-friendly [17,18].
HSO_5_^−^ + hv →SO_4_^•−^ + ·OH (1)
HSO_5_^−^ + heat →SO_4_^•−^ + ·OH(2)
HSO_5_^−^ + M^n+^ →SO_4_^•−^ + OH^−^ + M^n+1^(3)

Metal-organic Frameworks (MOFs) are a type of material with a periodic structure coordinated with inorganic metal nodes and organic bridge ligands. Their characteristics of rich topology, designability, high specific surface area, and easy functionality show great potential and wide application in many fields such as gas storage, separation, and pollutant removal [19,20]. In addition, the high density of unsaturated metal sites and large porosity in the MOF skeleton prevent the aggregation of active sites compared to the metal oxides [21,22]. Sun et al. [23] compared different iron-based MOFs including MIL-53(Fe), NH_2_-MIL-53(Fe), and Fe(BDC) (DMF, F) as activators of H_2_O_2_ for the degradation of phenol, and the Fe(BDC) (DMF, F)/H_2_O_2_ system showed the highest activity with 99% phenol removal and 78.7% TOC removal after 3 h. Lv et al. [24] synthesized MIL-100(Fe) for activation of H_2_O_2_ for dye wastewater treatment with an unsatisfied result, but the synthesis of Fe^II^@MIL-100(Fe) by optimization method greatly improved the dye wastewater treatment performance. These results showed that the unsaturated metal site Fe^II^ was the key to improving the catalytic activity of iron-containing MOFs. A similar conclusion was obtained from a study based on MOF activation of PDS to remove contaminants from water [18]. Pu et al. [25] also synthesized MIL-53(Fe) and used it to catalyze PDS for the removal of Orange G, which needed high PDS concentration and MIL-53(Fe) dosages. Hence, the synthesis of novel Fe-MOFs with unsaturated Fe^II^ to improve the catalytic activity and reduce the cost of wastewater treatment is urgently needed. It has been demonstrated that the introduction of another metal ion into MOFs could significantly enhance the catalytic performance [26]. Bimetallic FeCo-MOFs and CuCo-MOFs were synthesized and evaluated as catalysts of PMS for removal of organic pollutants in our previous studies [12,27,28]. It is well known that the structure, properties, and catalytic performance of MOFs are significantly influenced by different metal ions. In addition, copper and copper oxide also exhibited catalytic activity towards PMS [29]. Hence, it is necessary to evaluate the catalytic performance of bimetallic FeCu-MOFs for the activation of PMS.

In this study, 2,5-dihydroxyterephthalic acid (DHTA) was used as the organic ligand to synthesize the bimetallic metal-organic framework (MOF) FeCu-MOF by hydrothermal method. Scanning electron microscopy (SEM), transmission electron microscopy (TEM), X-ray powder diffraction (XRD), and Fourier transform infrared (FT-IR) spectroscopy were used to investigate the morphology, chemical composition, and textural properties of the as-prepared FeCu-MOF. The catalytic performance of the as-prepared FeCu-MOF was evaluated by the removal of methylene blue (MB). Furthermore, factors affecting the removal performance were investigated, including PMS concentration, catalyst dosage, initial pH, and common anions. The findings of this study will provide a new direction for the research and preparation of MOFs as heterogeneous catalysts for organic pollutant degradation. 

## 2. Materials and Methods

### 2.1. Materials

2,5-Dihydroxy-1,4-benzenedicarboxylate (DHTA), ferrous chloride tetrahydrate (FeCl_2_·4H_2_O), cupric chloride hydrate (Cu(NO_3_)_2_·3H_2_O), peroxydisulfate (Na_2_S_2_O_8_^2-^), and potassium monopersulfate triple salt (KHSO_5_**·**0.5KHSO_4_**·**0.5K_2_SO_4_) (purity > 47% KHSO_5_ basis) were provided by Macklin reagent Co., Ltd. (Shanghai, China), whereas N,N-dimethyl formamide (C_3_H_7_NO, DMF), methanol (MeOH), ethanol (EtOH), tert butyl alcohol (TBA), sodium nitrite (NaNO_2_), sodium sulfate (Na_2_SO_4_), sodium bicarbonate (NaHCO_3_), sodium hydrogen phosphate (Na_2_HPO_4_), sodium dihydrogen phosphate (NaH_2_PO_4_), sodium acetate (CH_3_COONa), methylene blue (MB, ≥90%, HPLC), and all other chemicals were purchased from Sinopharm Chemical Reagent Co., Ltd. (Beijing, China). Unless otherwise specified, all chemicals and reagents used in this study were of commercially available analytical grade and used without further purification. 

### 2.2. Preparation of FeCu-MOF

Bimetallic metal–organic frameworks (FeCu-MOFs) were synthesized through a solvothermal process referred to in our previous studies [27,28]. Briefly, 0.596 g (3.0 mM) of FeCl_2_·4H_2_O, 0.725 (3.0 mM) g of Cu(NO_3_)_2_·3H_2_O, and 0.594 (3.0 mM) of DHTA were dissolved in the mixed solutions of N,N’-dimethylformamide (DMF, 30 mL) and ethanol (EtOH, 30 mL) by magnetic stirring. Then the resulting mixture was transferred into a Teflon-lined stainless steel autoclave (100 mL) and heated in an oven at 150℃ for 24 h. After the mixture was cooled to room temperature, the products were obtained by filtration and alternately washed with DMF, methanol, and water. The products were dried under vacuum at 100 °C for 12 h. For comparison, the monometallic MOFs of Fe-MOF and Cu-MOF were also prepared under identical conditions. 

### 2.3. The Catalytic Performance of FeCu-MOF

The catalytic performance of as-prepared materials was investigated by the removal of MB. All batch experiments were carried out in a 250 mL conical flask, which was kept at a shaking rate of 120 rpm with a temperature of 25 °C in a water bath shaker (SHZ-C, Shanghai, China). Typically, 1.0 mL of PMS stock solution (200 mM) and 1.0 mL of MB stock solution (20 mM) were added into 98 mL of deionized water. Then the catalyst was added to initiate the oxidation reaction. At certain time intervals, samples were withdrawn and the residual MB concentration was analyzed using a 725 N UV-Vis spectrophotometer at 668 nm [9,24]. The initial pH was adjusted using 1.0 mol/L sodium hydroxide (NaOH) or sulfuric acid (H_2_SO_4_). 

The MB removal process was fitted to the pseudo-first-order kinetics, where *C*_0_ (mM) and *C_t_* are the initial and after time *t* (min) concentrations of MB during the oxidation process, respectively, and *k*_obs_ is the observed pseudo-first-order constant (min^−1^).
(4)lnctc0=−kobst

### 2.4. Analytical Methods

The morphological observation of the as-prepared catalyst was performed using a scanning electron microscope (ZEISS Ultra 55, Jena, Germany) and transmission electron microscope (FEI Tecnai F30, Hillsboro, OR, USA). The catalyst phase and crystal structure of catalysts were determined using an X-ray diffractometer (D8 Advance X-ray Diffraction system, Bruker AXS D8 Advance, Karlsruhe, Germany). The metal valance state was identified by X-ray photoelectron spectroscopy (XPS, Thermo Fisher Scientific K-Alpha, Waltham, MA, USA). The Brunauer–Emmett–Teller (BET) specific surface area (SBET) analysis was performed on an ASAP 2010 surface area analyzer (Micromeritics, Norcross, GA, USA). The amount of iron ions leaching in water was measured with 1,10-phenanthroline at a wavelength of 510 nm [30]. The degradation products of MB were analyzed by UPLC/MS (thermo scientific Q Exactive). A capillary column Accucore aQ (150 mm × 2.1 mm × 2.6 μm) was used for the separation of intermediates. The detection system was a diode array detector (Thermo Scientific, Berlin, Germany) with a detection range between 190 and 800 nm. The mobile phase consisted of two solutions, namely A (0.1% acetic acid (pH 5.3)) and B (acetonitrile). The gradient UPLC separation was coupled with an ion trap mass spectrometer (Agilent Technologies). The mass spectrometer was equipped with an electrospray ionization source and operated in positive polarity. The ESI conditions were set as follows: spray voltage: 3.2 kV, capillary temperature: 300.00 °C; sheath gas: 40.00; aux gas: 15.00; max spray current: 100.00; probe heater temperature: 350.00 °C; S-Lens RF level: 50.00.

## 3. Results and Discussion

### 3.1. Characterization of FeCu-MOF

#### 3.1.1. Crystallographic Structure

Figure 1a shows the XRD patterns of the as-prepared Fe-MOF, Cu-MOF, and FeCu-MOF. The Fe-MOFs constructed by self-assembly of ferrous chloride and DHTA via hydrothermal showed a low crystallinity, with diffraction peaks at 2θ = 11.96°, 15.89°, 16.9°, 26.7°, and 35.2°. Similar diffraction peaks were observed in XRD patterns of Cu-MOF. However, the intensity of these diffraction peaks was much greater than that of the diffraction peaks of Fe-MOF, indicating that Cu-MOF has a higher degree of crystallinity. The characteristic peaks that appeared in the bimetallic FeCu-MOF were the weakest. This may be due to the fact that the chelating ability between iron and DHTA was higher than that between copper ions and DHTA and led to the formation of Fe-MOF with low crystallinity.

Fourier-transform infrared (FT-IR) spectroscopy was used to detect the surface functional groups in the as-prepared materials, as seen in Figure 1b. The absorption peaks at 1650 cm^−1^ and 1427 cm^−1^ in the ligands of DHTA samples were mainly due to v_s_(-COO-) and v_as_(-COO-) of coordination bonds between the organic ligands and the hydrogen atoms [26]. The absorption peaks at 750 cm^−1^ and 590 cm^−1^ were ascribed to the vibration of the C-H bond in the benzene ring [31,32]. Similar vibration absorption peaks were found in the infrared spectra of Fe-MOF, Cu-MOF, and FeCu-MOF, but most of these peaks shifted after the coordination reaction between metal ions and organic ligands [27,33]. This result indicated that metal ions and DHTA were successfully coordinated to form skeleton materials.

Surface properties are crucial to the catalytic performance of materials [34]. Brunauer-Emmett-Teller (BET) gas sorption analysis was performed o characterize the porous structure and surface area of the bimetallic FeCu-MOF, and the resulting curves are shown in Figure 1c,d. It can be seen that FeCu-MOF showed type IV isotherms with mesoporous specific surface areas of 6.13 m^2^/g, indicating the presence of a mesoporous structure in these samples. The presence of a mesoporous structure could facilitate the mass transportation between oxidants and pollutants [35,36]. The BJH adsorption average pore diameter was 20.03 nm (Figure 1d), which further indicated the existence of a mesoporous structure in FeCu-MOF. The low BET surface area was due to the fact that the microporous specific surface areas were not detected. On the other hand, the introduction of a second metal ion may enlarge the pore structure and lead to a low BET; a similar result was found in a previous study [12].

#### 3.1.2. Morphology Characterization

Figure 2a displays SEM images of FeCu-MOF, which displayed a hydrangea-like framework with a size of 2 μm. Evidently, the hydrangea-like framework is composed of flake crystals with a width of 200–500 nm and a length of 1 μm (Figure 2b). The observation of the TEM pattern of FeCu-MOF shown in Figure 2c further revealed the hydrangea-like structure. In addition, a large number of nanoparticles were observed on the flake crystal, as shown in Figure 2d,e. Besides the hydrangea-like structure, a rod-like structure with a length of 1.0–2.0 μm was also observed (Figure 2f). Nanoparticles were also found on the rod-like structure. According to the HRTEM of FeCu-MOF, the lattice spacings of 0.2427 and 0.2130 nm were matched well with the reflection facets (111) and (200) of Cu_2_O (JCPDS 34-1354), respectively. These results revealed that the formed nanoparticles were most likely Cu_2_O which may be reduced by DMF during the hydrothermal process [3].

### 3.2. Catalytic Performance

The catalytic activity of the bimetallic FeCu-MOF towards PMS was evaluated by determining the removal rates of MB. As shown in Figure 3, 43.4% of MB was removed in 60 min by solo oxidation of PMS. The removal rate of MB was less than 13% in the presence of only FeCu-MOF, revealing that FeCu-MOF showed weak adsorption capacity for MB. However, the removal rate of MB was 70.1% in the presence of both PMS and FeCu-MOF, indicating that FeCu-MOF exhibited good catalytic activity towards PMS. As a comparison, MB removal experiments were also performed in bimetallic FeCu-MOF/PDS and monometallic MOF/PMS systems. It was clearly seen that 40% of MB was removed at a rapid speed in the first 20 min in the FeCu-MOF/PDS system. However, the degradation process seemed to stop when the reaction time was prolonged. The removal rates of MB were 52.8% and 38.2% in the Fe-MOF/PMS and Cu-MOF/PMS systems, respectively, which were much lower than that in FeCu-MOF/PMS system. The low MB removal efficiency in Cu-MOF/PMS was probably due to the low catalytic performance of Cu-MOF towards PMS and the PMS adsorption on the catalyst, which hindered the MB oxidation and resulted in the MB removal being lower than that observed for solo PMS oxidation [37].

### 3.3. Effects of Preparation Condition

#### 3.3.1. Effect of PMS Concentration and Catalyst Dosage

The effects of different PMS concentrations (1.0–6 mM) on the MB removal were investigated (Figure 4a). The result showed that the MB removal rate increased gradually with the increase in PMS concentration, ranging from 58.1% after 60 min with 1.0 mM to 100% with 6.0 mM. As shown in Figure 4a and Table 1, the MB removal exhibited a good fit to a pseudo-first-order model, and all R^2^ values were greater than 0.95. The rate constant (*k*_obs_) of MB removal increased from 0.014 min^−1^ to 0.099 min^−1^ as the PMS concentration increased from 1.0 mM to 6.0 mM. This result indicated that the removal of MB in this system strongly relied on the PMS concentration. As seen in Figure 4b, the removal rate of MB increased with the increase in FeCu-MOF dosage. The *k*_obs_ of MB removal increased from 0.019 min^−1^ to 0.025 min^−1^, 0.032 min^−1^, 0.037 min^−1^, and 0.062 min^−1^ when the FeCu-MOF dosage increased from 0.05 g/L to 0.1 g/L, 0.2 g/L, 0.3 g/L, and 0.6 g/L, respectively. The increase in removal rate was due to the fact that the increased catalyst could effectively increase the number of active sites for activation of PMS, which can accelerate the generation of reactive oxygen species (ROS) for oxidation of MB.

#### 3.3.2. Effect of Initial MB Concentration and pH

As displayed in Figure 5a, MB removal efficiency gradually decreased with the increase in initial MB concentration. Correspondingly, the *k*_obs_ of MB removal decreased from 0.025 min^−1^ to 0.0048 min^−1^ when the MB concentration increased from 0.1 mM to 1.2 mM; similar results were found in other catalyst-based oxidation processes [12]. The solution pH plays an important role in organic pollutant removal in PMS-based oxidation systems, as it can influence the speciation of PMS (Equations (5) and (6)) [38]. It is necessary to investigate the effect of solution pH on the removal of MB. The effect of solution pH ranging from 3.16 to 10.87 on MB removal was explored, and the result is displayed in Figure 5b.
HSO_5_^−^ ↔ SO_5_^2^^−^ + H^+^ pK_a_ = 9.4(5)
HSO_5_^−^ + OH^−^ → SO_5_^2^^−^ + H_2_O(6)
Alkaline pH: SO_4_^•^^−^ + OH^−^ → SO_4_^2^^−^ + **·**OH *k* = 6.5 × 10^7^ M^−1^s^−1^
(7)

A homogeneous Fenton reaction usually achieves the best pollutant removal efficiency in the pH range of 3.0−5.0 [39]. Similarly, when the initial pH gradually increased from 3.16 to 6.96, the *k*_obs_ of MB removal gradually decreased from 0.0215 min^−1^ to 0.0143 min^−1^. However, when the initial pH was further increased to 9.05, the *k*_obs_ increased to 0.0203 min^−1^. The largest *k*_obs_ was obtained at pH 3.16, while the smallest *k*_obs_ (0.0086 min^−1^) was obtained at pH 10.87. The rapid decline in MB removal was probably due to the translation from SO_4_^•^^−^ to **·**OH (Equation (7)) [38]. On the other hand, the effective ferrous iron ions form iron sludge and lose catalytic performance under strongly alkaline conditions [39]. These results indicated that acidic conditions were favored for MB removal in the FeCu-MOF/PMS system.

As displayed in Table 2, the catalytic performance of FeCu-MOF was also compared with that of other previously reported MOF catalysts for the removal of dyes. According to the dye removal efficiencies and reaction conditions, PMS activated with FeCu-MOF in this study showed superior or comparable catalytic ability to the other processes. 

#### 3.3.3. Effect of Common Anions and Radical Scavengers 

It is well known that industrial wastewater and surface water usually contain many anions that can affect the treatment of wastewater in advanced oxidation systems [40]. In order to understand the effect of common anions on the removal of MB in the FeCu-MOF/PMS system, the effects of different anions with concentrations of 50 mM on the MB removal were investigated. As shown in Figure 6a, most of the anions displayed inhibition effects on the MB removal. It was seen that the nitrite ion (NO_2_^−^) showed the strongest inhibition effect on MB with a *k*_obs_ of 0.0008 min^−1^, followed by H_2_PO_4_^−^ (0.0114 min^−1^), SO_4_^2^^−^ (0.014 min^−1^), HPO_4_^2^^−^ (0.0152 min^−1^), and HCO_3_^−^ (0.0173 min^−1^). On the contrary, the addition of acetate ion (CHCOO^−^) promoted the removal of MB and achieved a *k*_obs_ of 0.026 min^−1^. This was probably due to the fact that CHCOO^−^ can act as an electron shuttle to promote electron transfer, thus improving the catalytic activity of PMS [20]. 

SO_4_^•^^−^ and **·**OH are usually identified as major reactive species for oxidation of pollutants in PMS-AOP systems [41,42,43]. Hence, it is necessary to identify the generated reactive species that are responsible for the MB removal in the FeCu-MOF/PMS system. Radical scavengers have been successfully used in identifying the generation of ROS due to the different reaction rates between scavengers and ROS [5,7]. In this study, methanol (MeOH) was used to capture both SO_4_^•^^−^ and **·**OH, while tert-butyl alcohol (TBA) was applied to capture **·**OH. As shown in Figure 6b, the *k*_obs_ of MB removal decreased from 0.025 min^−1^ to 0.022 min^−1^ and 0.013 min^−1^ with the addition of TBA and MeOH, respectively. This result revealed that the MB removal was mainly due to the oxidation of SO_4_^•^^−^ [44,45]. In addition, the addition of His also resulted in a strong inhibition of MB removal, indicating the generation of ^1^O_2_ in the FeCu-MOF/PMS system.

### 3.4. Reusability of FeCu-MOF 

To evaluate the reusability of FeCu-MOF, recycling runs using the same FeCu-MOF were performed under the optimal reaction conditions. After each reaction cycle, the used catalyst was collected and washed three times. As displayed in Figure 7, the MB removal efficiency only decreased a little compared with the first run, where 100%, 97.1%, and 87.1% of MB were removed within 60 min in the first, second, and third runs, respectively. The slight decline in the MB removal efficiency was probably due to the loss of FeCu-MOF, which adsorbed on the filter paper and leached into aqueous solutions during the recycling process. This result indicated that efforts are still urgently needed to improve the water stability of FeCu-MOF from the viewpoint of engineering application.

### 3.5. Possible Mechanism of PMS Activation by FeCu-MOF and MB Removal Mechanism

On the basis of the foregoing discussion, MB removal in the FeCu-MOF/PMS system was ascribed to the generation of SO_4_^•^^−^, **·**OH, and ^1^O_2_. To clarify the activation mechanism and understand the formation of these ROS, XPS was investigated before and after the reaction. The XPS survey spectra (data not shown) clearly showed the existence of Fe (~5.35 at%), Cu (~0.22 at%), C (~59.29 at%), and O (~35.13 at%) elements in fresh FeCu-MOF. Although the feeding molar ratio of Fe^2+^/Cu^2+^ was set to 1/1, the XPS-determined molar ratio of Fe^2+^/Cu^2+^ was found to be 24.3/1. This result revealed that only a small part of Cu^2+^ took part in the synthesis reaction during the hydrothermal process. According to the SEM and TEM characterization, the Cu^2+^ was reduced to form Cu_2_O. 

The high-resolution C 1s is shown in Figure 8a; the two peaks at 284.8 eV and 286.1 eV were respectively ascribed to the C-C/C-H bonds and C=O bonds in organic ligands. The peak located at 288.3 eV corresponded to carboxyl COOH bonds in ligands [26,27,33]. According to the O 1s high-resolution XPS spectrum of FeCu-MOF (Figure 8b), the O 1s peak was fitted into two peaks at 531.1 eV and 532.1 eV, which corresponded to Fe/Cu-OH and O-C=O [46]. As seen in Figure 8c, the Fe 2p_3/2_ spectra can be deconvoluted into three peaks centered at 709.7 eV, 711.4 eV, and 714.6 eV. Those peaks at 709.7 eV and 714.6 eV were assigned to the characteristic peaks of Fe(II), while the peak at 711.4 eV corresponded to Fe(III). After the reaction, the two peaks ascribed to Fe(II) shifted to 710.7 eV and 715.2 eV (see Table 3), respectively. Moreover, the peak area significantly decreased, indicating that Fe(II) participated in the PMS activation [43] (Equations (8) and (9)). Due to the low loading in the catalyst, the Cu 2p high-resolution XPS spectrum showed a small peak located at 932.7 eV (Figure 8d), which was assigned to Cu(I). The formation of Cu(I) caused a series of reactions (Equations (10)–(13)) and generated ^1^O_2_ to oxidize MB. The generation of a small amount of **·**OH was mainly due to the transformation from SO_4_^•^^−^ (Equations (3) and (14)). In addition, the shift of characteristic peaks of metal species (see Table 3) after the reaction further indicated that both Fe(II) and Cu(I) participated in activating PMS [47,48].
≡Fe(II) + HSO_5_^−^ → ≡Fe(III) + SO_4_^•^^−^ + OH^−^(8)
≡Fe(III) + HSO_5_^−^ → ≡ Fe(II)) + SO_5_^•^^−^ + H^+^(9)
≡Cu(I) + HSO_5_^−^ → ≡Cu(II) + OH^−^ + SO_4_^•−^(10)
≡Cu(II) + 2HSO_5_^−^ + H_2_O → ≡Cu(I) + O_2_^•−^+ S_2_O_8_^2−^ + 2H^+^(11)
2O_2_^•−^+ 2H_2_O → H_2_O_2_ + 2OH^−^ + ^1^O_2_(12)
2**·**OH + H_2_O_2_ → ^1^O_2_ + 2H_2_O(13)
Aqueous solutions: SO_4_^•^^−^ + H_2_O → SO_4_^2-^ + **·**OH + H^+^ *k* < 1.0 × 10^3^ M^−1^s^−1^(14)

In order to determine the contribution of the homogeneous reaction to the MB removal in the FeCu-MOF/PMS system, the variation of ferrous and total iron ion concentrations under different pH values was detected. As shown in Figure 9a, the ferrous iron concentration at pH 3.16 was 4.2 mg/L at 10 min, but it decreased to 3.2 mg/L and then increased to 4.75 mg/L when the reaction time was increased to 30 and 60 min, respectively. The decrease in ferrous iron concentration may be due to PMS oxidation (Equation (15)), and the increase in total iron ions (Figure 9b) further confirmed this inference. On the contrary, the concentration of ferrous iron at pH 5.33 was increased firstly and then decreased, but the concentration of total iron ions also increased with the increase in the reaction time. Under neutral and alkaline conditions (pH= 6.96, 10.87), the concentration of ferrous iron increased with the increase in time, as did the total iron concentration. At pH 10.87, the concentration of ferrous iron reached 4.86 mg/L at 60 min. All these results indicated that the homogeneous reaction also played a nonnegligible role in the MB removal (Equations (15)–(17)).
Fe^2+^ + HSO_5_^−^→Fe^3+^ + OH^−^ + SO_4_^•^^−^ *k* = 3.0.0 × 10^4^ M^−1^s^−1^(15)
Fe^3+^ + HSO_5_^−^→Fe^2+^ + SO_5_^•^^−^ + H^+^(16)
2SO_5_^•^^−^ + H_2_O → 2 HSO_4_^−^ +1.5^1^O_2_(17)

According to the result of UPLC-MS, four intermediates of MB were identified after 60 min reaction in the FeCu-MOF/PMS system: azure A (*m*/*z* = 256), thionin (*m*/*z* = 228), 4-aminobipheyl (*m*/*z* = 169), and 1-phenylethanamine (*m*/*z* = 121). All these phenolic compounds revealed that further oxidation was needed to oxidize these intermediates into small molecular substances that finally mineralize to CO_2_ and H_2_O. Based on the foregoing discussion, a possible activation mechanism for oxidation of MB is described in Figure 1.

## 4. Conclusions

This work demonstrated a strategy for the synthesis of bimetallic MOF catalysts for effective heterogeneous activation of PMS to remove organic contaminants from water. After the introduction of the copper ion into Fe-MOF, the obtained bimetallic FeCu-MOF exhibited remarkable efficiency compared with either Fe-MOF or Cu-MOF. In the presence of 0.6 g/L FeCu-MOF and 6.0 mM PMS, 64 ppm MB (0.2 mM) can be 100% removed in 30 min. XRD, FT-IR, SEM, TEM, BET, and XPS were applied to characterize the physicochemical properties of the as-synthesized bimetallic MOFs which displayed the hydrangea-like and rod-like structures with a homogeneous distribution of unsaturated metal sites and an abundant mesoporous structure. Effects of FeCu-MOF dosage, PMS concentration, initial MB concentration, initial pH, and common inorganic ions on the removal of MB were investigated. The results indicated that MB removal increased with the increase in the FeCu-MOF dosage and PMS concentration, but decreased with the increase in the initial MB concentration. Besides CHCOO^−^, other anions presented evident inhibitory effects on the removal of MB, which decreased in the order NO_2_^−^ (0.0008 min^−1^) > H_2_PO_4_^−^ (0.0114 min^−1^) > SO_4_^2−^ (0.014 min^−1^) > HPO_4_^2−^ (0.0152 min^−1^) > HCO_3_^−^ (0.0173 min^−1^). The effects of initial pH on the MB removal were ranked as follows: pH 3.16 > pH 9.05 > pH 5.33 > pH 6.96 > pH 10.87. Although SO_4_^•^^−^, **·**OH, and ^1^O_2_ coexisted in the FeCu-MOF/PMS system, SO_4_^•^^−^ acted as the predominant reactive species for MB removal. The possible mechanism based on both heterogeneous and homogeneous activation of PMS was proposed, along with the MB oxidation mechanism. Regarding the MB intermediates, the catalytic performance needs to improve and mineralize the phenolic intermediates of MB to CO_2_ and H_2_O. From the viewpoint of engineering application, efforts are still needed to improve the water stability and reusability of FeCu-MOF in future studies.

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
