# Peer review of "Synthesis of Bimetallic FeCu-MOF and Its Performance as Catalyst of Peroxymonosulfate for Degradation of Methylene Blue"

_materials, 2022, doi:10.3390/ma15207252_

Round 1
Reviewer 1 Report
Review Report for Materials
Synthesis of bimetallic FeCu-MOF and the performance as cat- 2 alyst of peroxymonosulfate for degradation of methylene blue
My Queries:
- Authors have to present the full form of PMS and PDS in the abstract otherwise it is confusing for the readers.
- Authors have mentioned Fig.1d, but there is not Fig. 1d in the manuscript.
- Authors should be cited the relevant MOF citations for the FT-IR compared to present research data such as Appl. Sci. 2021, 11(21), 9856; https://www.mdpi.com/2076-3417/11/21/9856.
- I recommend authors to show how methylene blue(MB) was oxidized though bimetallic MOF by mechanistic aspect including chemical structure of MB with respect to Fig. 5, it should be helpful to readers.
- I don’t find the 1, 10-phenanthorline data of iron ions leaching in the manuscript.
- In the FI-IR, Fe-MOF and Cu-MOF are looking similar graphs, there is no difference btw them.
- Line no. 217, stated that, 52.8% for Fe-MOF/PMS, and 38.2% for Cu-MOF/PMS, respectively, however, there is no explanation why it was showed poor removal rate, need cited reference to support your research.
- Table 1, 0.2mM of 0.1g/L, 2.0 mM at pH~10.87, MB observation is decreased to 0.0086, what is the reason? Any citations.
- Table 2, the binding energies of each element of Fe and Cu are well matched before and after reaction without any changes, it means the respective ions are presenting in the MOFs. It’s a positive comment but need exact citation.
- Scheme 2, authors have shown homogeneous activation but how MB can change mechanistically need to show.
- At lease need to provide Fe/Cu-MOF and FeCu-MOF’s 1HNMR data.
- Using these MOFs, can they reuse for MB removal? No information.
Decision: If the above amendments are made, I hope it can be accepted.
Prof. Ravi Kumar Cheedarala,
Mechanical Engineering,
Changwon National University,
Changwon,
S.Korea
·

Author Response
Response to Reviewer #1
We very much appreciate the careful reading of our manuscript and valuable suggestions of the reviewer. We have carefully considered the comments and have revised the manuscript accordingly. Below is our point-by-point response to the comments of Reviewer #1.
- Authors have to present the full form of PMS and PDS in the abstract otherwise it is confusing for the readers.
Response: Thanks very much for the comments. As suggested by the reviewer, we have presented the full form of PMS and PDS in the abstract, please see in Page 14.
- Authors have mentioned Fig.1d, but there is not Fig. 1d in the manuscript.
Response: Thanks very much for the comments. We have added the number in Fig.1d which was inset into Fig. 1c, please see Fig. 1 in page 5.
- Authors should be cited the relevant MOF citations for the FT-IR compared to present research data such as Appl. Sci. 2021, 11(21), 9856; https://www.mdpi.com/2076-3417/11/21/9856.
Response: Thanks very much for the comments. As suggested by the reviewer, we have cited the relevant research for the FT-IR analysis. Please see in Line 175 in the Page 4.
- I recommend authors to show how methylene blue(MB) was oxidized though bimetallic MOF by mechanistic aspect including chemical structure of MB with respect to Fig. 5, it should be helpful to readers.
Response: Thanks very much for the comments. As suggested by the reviewer, the degradation products of MB and the degradation pathway was added in Scheme 2. Please see in Line 175 in the Page 13, 14.
According to the result of UPLC-MS, four intermediates of MB were identified after 60 min reaction in the FeCu-MOF/PMS system, which includes azure A (m/z=256), thionin (m/z=228), 4-Aminobipheyl (m/z=169), and 1-Phenylethanamine (m/z=121). All these phenolic compounds revealed that further oxidation was needed to oxidize these intermediates into small molecular substances and finally mineralize to CO2 and H2O. Based on the foregoing discussion, the possible activation mechanism of PMS for oxidation of MB was described in Scheme 2.
Scheme 2. The possible PMS activation mechanism by FeCu-MOF and MB degradation mechanism.
- I don’t find the 1, 10-phenanthorline data of iron ions leaching in the manuscript.
Response: Thanks very much for the comments. The 1, 10-phenanthorline data of iron ions leaching was displayed in Fig. 9 of page 14.
Please see our revised manuscript (P. 13, 14):
In order to clear the contribution of homogeneous reaction to the MB removal in the FeCu-MOF/PMS system, the variation of ferrous and total iron ions concentrations under different pH values was detected. As shown in Fig. 9a, the ferrous iron concentration at pH 3.16 was 4.2 mg/L at 10 min, but it reduced to 3.2 mg/L and then increased to 4.75 mg/L when the reaction time was increased to 30 and 60 min, respectively. The decrease of ferrous iron concentration may be due to the PMS oxidation (Eq. (15)), the increase of total iron ions (Fig. 9b) further confirmed this inference. On the contrary, the concentration of ferrous iron at pH 5.33 was increased firstly and then decreased, but the concentration of total iron ions also increased with prolong the reaction time. Under neutral and alkaline conditions (pH= 6.96, 10.87), the concentration of ferrous iron increased with the increase of time, as well as the total iron concentrations. At pH 10.87, the concentration of ferrous iron reached 4.86 mg/L at 60 min. All these results indicated that homogeneous reaction also played a nonnegligible role in the MB removal (Eqs. (15)- (17)).
Fe2+ + HSO5−→Fe3+ + OH−+ SO4•− k = 3.0.0 × 104 M-1s-1 (15)
Fe3+ + HSO5−→Fe2+ + SO5•− + H+ (16)
2SO5•− + H2O → 2 HSO4− +1.51O2 (17)
Figure 9. (a) The variation of ferrous iron ions under different initial pHs; (b) The variation of total iron irons under different initial pHs.
- In the FI-IR, Fe-MOF and Cu-MOF are looking similar graphs, there is no difference btw them.
Response: That is right. As the FI-IR was mainly used to detect the surface functional groups in the as-prepared materials, Fe-MOF and Cu-MOF contain the same organic ligand of DHTA, which would display similar FI-IR.
- Line no. 217, stated that, 52.8% for Fe-MOF/PMS, and 38.2% for Cu-MOF/PMS, respectively, however, there is no explanation why it was showed poor removal rate, need cited reference to support your research.
Response: Thanks very much for the comments. As suggested by the reviewer, explanation for the poor removal rate and relevant reference were added to support our research.
Please see our revised manuscript (Page 7):
The low MB removal efficiency in Cu-MOF/PMS was probably due to the low catalytic performance of Cu-MOF towards PMS and the PMS adsorption on the catalyst, which hindered the MB oxidation and resulted in the lower MB removal than solo PMS oxidation [35].
- Table 1, 0.2mM of 0.1g/L, 2.0 mM at pH~10.87, MB observation is decreased to 0.0086, what is the reason? Any citations.
Response: Thanks very much for the comments. As suggested by the reviewer, explanation for the decline in MB removal rate and relevant reference were added in our revised manuscript.
Please see our revised manuscript (Page 8):
The largest kobs was obtained at pH 3.16 while the smallest kobs (0.0086 min-1) was obtained at pH 10.87. The rapid decline in MB removal was probably ascribe to the translation from SO4•− to .OH (Eq. (7)) [36]. On the other hand, the effective ferrous iron ion would form iron sludge and lose catalytic performance under strong alkaline conditions [37].
- Table 2, the binding energies of each element of Fe and Cu are well matched before and after reaction without any changes, it means the respective ions are presenting in the MOFs. It’s a positive comment but need exact citation.
Response: Thanks very much for the comments. As suggested by the reviewer, relevant reference was added in our revised manuscript, please see Page 12, 13.
- Scheme 2, authors have shown homogeneous activation but how MB can change mechanistically need to show.
Response: Thanks very much for the comments. As suggested by the reviewer, MB degradation mechanism was added into Scheme 2.
Please see in Line 175 in the Page 13, 14.
According to the result of UPLC-MS, four intermediates of MB were identified after 60 min reaction in the FeCu-MOF/PMS system, which includes azure A (m/z=256), thionin (m/z=228), 4-Aminobipheyl (m/z=169), and 1-Phenylethanamine (m/z=121). All these phenolic compounds revealed that further oxidation was needed to oxidize these intermediates into small molecular substances and finally mineralize to CO2 and H2O. Based on the foregoing discussion, the possible activation mechanism of PMS for oxidation of MB was described in Scheme 2.
Scheme 2. The possible PMS activation mechanism by FeCu-MOF and MB degradation mechanism.
- At lease need to provide Fe/Cu-MOF and FeCu-MOF’s 1HNMR data.
Response: Thanks very much for the comments. However, to the best of our knowledge, the 1HNMR data could not useful information for the catalytic performance of MOFs in this system. The 1HNMR data is appropriate for the completely new materials. The MOFs synthesis in this study is not completely new but its use is new, hence, there is no need to provide 1HNMR data. Many similar studies did not provide 1HNMR data, for example “Ferrous metal-organic frameworks with stronger coordinatively unsaturated metal sites for persulfate activation to effectively degrade dibutyl phthalate in wastewater, Journal of Hazardous Materials 377 (2019) 163–171; Synthesis of iron-based metal-organic framework MIL-53 as an efficient catalyst to activate persulfate for the degradation of Orange G in aqueous solution, Applied Catalysis A, General 549 (2018) 82–92” and so on”.
- Using these MOFs, can they reuse for MB removal? No information.
Response: Thanks very much for the comments. As suggested by the reviewer, the reusability of FeCu-MOF was added into our revised manuscript.
Please see in Line 175 in the Page 11.
Figure 7. Reusability of FeCu-MOF for MB removal. Reaction conditions: MB = 0.2 mM; PMS = 6.0 mM; FeCu-MOF = 0.6 g/L; pH unadjusted.
To evaluate the reusability of FeCu-MOF, recycling runs using the same FeCu-MOF were performed under the optimal reaction conditions. After each reaction cycle, the used catalyst was collected and washed three times. As displayed in Figure 7, the MB removal efficiency only decreased a little compared with the first run, where 100%, 97.1% and 87.1% of MB were removed within 60 min in the 1st, 2nd, and 3rd runs, respectively.The slight decline in the MB removal efficiency was propably due to the loss of FeCu-MOF, which adsorbed on the filter paper and leached into aqueous solutions during the recycling process. This result indicated that efforts are still urgently needed to improving the water stability of FeCu-MOF from the viewpoint of engineering application.

Reviewer 2 Report
The manuscript is interesting, the authors have reported the roles of bimetallic Fe-Cu MOF in the degradation of MB dye. I recommend the acceptance of this manuscript after minor revision.
1. The authors are requested to calculate the band gap values of the Fe-Cu MOF.
2. The degradation of MB is reported, however, monitored UV-Vis spectroscopical graphs are missing.
3. The photocurrent and EIS studies are missing.
4. It is suggested that the authors can clarify the novelty of the manuscript further by comparison with the reported papers.
5. The product from the photodegradation of MB should be studied by GC-MS and discussed in the revised manuscript.
6. In the section introduction, the advances in photocatalysis should be improved, and the following articles are recommended to be cited
Appl. Surf. Sci. 486, 198–211; Appl. Catal. B Environ. 235, 225–237; J. Photochem. Photobiol. A Chem. 368, 168–181; Mater. Lett. 108, 72–74; J. Environ. Manage. 256, 109930.
7. Moderate English (language and style) improvement is required.
8. Please mention some possible disadvantages of your proposed method for decontamination of MB pollutants. (maybe in the conclusion part).
Author Response
Response to Reviewer #2
We very much appreciate the careful reading of our manuscript and valuable suggestions of the reviewer. Thank you for the kind advice. We have carefully considered the comments and have revised the manuscript accordingly. Below is our point-by-point response to the comments of Reviewer #2.
- The authors are requested to calculate the band gap values of the Fe-Cu MOF.
Response: Thanks for the Reviewer’s comment. This study was aim to synthesis of effective catalyst to activate PMS for oxidation organic pollutants. Band gap values of materials were important for the photocatalysts. The catalyst used in this study is not photocatalysts. The reaction mechanism is similar to photodegradation of organic pollutants, but this system does not involved photo. Hence, there is no need to calculate the band gap values of the Fe-Cu MOF.
- The degradation of MB is reported, however, monitored UV-Vis spectroscopical graphs are missing.
Response: The degradation of MB was reported by many researches, the detection of MB and other organic dyes concentration by UV-Vis spectrophotometer become more and more mature and have been used widely due to the fact it is convenient and simple. In order to save the cost and resource, there is no need to monitor UV-Vis spectroscopical graphs. Instead, we have cited two studies to support the detection method of MB, please see our revised manuscript in Line 132. In addition, many studies that reported dyes degradation also did not monitored UV-Vis spectroscopical graphs, including “H.L.Z. Lv, H. Y.; Cao, T. C.; Qian, L.; Wang, Y. B.; Zhao, G. H., Efficient degradation of high concentration azo-dye wastewater by heterogeneous Fenton process with iron-based metal-organic framework, J. Mol. Catal. A-Chem., 400 (2015) 81-89”,” H. Li, J. Zhang, Y. Yao, X. Miao, J. Chen, J. Tang, Nanoporous bimetallic metal-organic framework (FeCo-BDC) as a novel catalyst for efficient removal of organic contaminants, Environmental pollution, 255 (2019) 113337”, “ H.L.Z. Lv, H. Y.; Cao, T. C.; Qian, L.; Wang, Y. B.; Zhao, G. H., Efficient degradation of high concentration azo-dye wastewater by heterogeneous Fenton process with iron-based metal-organic framework, J. Mol. Catal. A-Chem., 400 (2015) 81-89.”
- The photocurrent and EIS studies are missing.
Response: Thanks for the comments. This study was aim to synthesis of effective catalyst to activate PMS for oxidation organic pollutants. photocurrent and EIS studies of materials were important for the photocatalysts. The catalyst used in this study is not photocatalysts. The reaction mechanism is similar to photodegradation of organic pollutants, but this system does not involved photo. Photocurrent and EIS studies for this system could not provide useful information for the catalytic performance and the MB degradation mechanism. In order to save the cost and resource, there is no need to investigate photocurrent and EIS of the Fe-Cu MOF.
- It is suggested that the authors can clarify the novelty of the manuscript further by comparison with the reported papers.
Response: Thanks very much for the comments. As suggested by the reviewer, we have presented the comparison with the reported papers to clarify the novelty of the manuscript in our revised manuscript, please see in Pages 8, 9.
Table 2 Comparative results of dyes removal by various methods.
|
Type of catalyst |
Reaction system |
Reaction conditions |
Performance |
References |
|
MIL-53(Fe) |
PDS |
OG=0.2 mM, Catalyst=1.0 g/ L, PDS= 32 mM |
120 min, 100% |
[23] |
|
MOF/carbon aerogel |
UV=500W |
RhB =50 mg/L, Catalyst=3.0g/ L, pH = 3.0 |
45 min, 100% |
[24] |
|
Cu@Co-MOFs |
PMS |
MB=0.2 mM, Catalyst=0.1 g/ L, PMS= 2 mM |
30 min, 100.0% |
[3] |
|
Cu@Co-MOFs |
PDS |
MB=0.2 mM, Catalyst=0.1 g/ L, PDS= 2 mM |
30 min, 28.0% |
[3] |
|
MIL-100(Fe) |
H2O2 |
MB = 0.5 g/L, catalyst = 1.0 g/L, H2O2 = 40 mM, pH = 3.0 |
285 min, 45% |
[22] |
|
FeII@MIL-100(Fe) |
H2O2 |
MB = 0.5 g/L, catalyst = 1.0 g/L, H2O2 = 40 mM, pH = 3.0 |
285 min, 78% |
[22] |
|
FeCo-BDC |
PMS |
MB=0.2 mM, catalyst = 50 mg/L, PMS = 1.0 mM |
15 min, 100% |
[25] |
|
CuCo-MOF-74 |
PMS |
MB=0.2 mM, catalyst = 50 mg/L, PMS = 2.0 mM |
30 min, 100.0% |
[12] |
|
FeCu-MOF |
PMS |
MB=0.2 mM, Catalyst=0.6 g/L, PMS= 6.0 mM, |
30 min, 100% |
This study |
As displayed in Table 2, the catalytic performance of FeCu-MOF was also compared with other previously reported MOFs catalysts for removal of dyes. According to the dyes removal efficiencies and their reaction conditions, PMS activated with FeCu-MOF in this study showed superior or comparative catalytic ability to the other processes.
- The product from the photodegradation of MB should be studied by GC-MS and discussed in the revised manuscript.
Response: Thank the reviewer for the comments. As suggested by the reviewer, the degradation products of MB was studied by HPLC-MS and the degradation pathway was added in Scheme 2. Please see in Line 175 in the Page 13, 14.
According to the result of UPLC-MS, four intermediates of MB were identified after 60 min reaction in the FeCu-MOF/PMS system, which includes azure A (m/z=256), thionin (m/z=228), 4-Aminobipheyl (m/z=169), and 1-Phenylethanamine (m/z=121). All these phenolic compounds revealed that further oxidation was needed to oxidize these intermediates into small molecular substances and finally mineralize to CO2 and H2O. Based on the foregoing discussion, the possible activation mechanism of PMS for oxidation of MB was described in Scheme 2.
Scheme 2. The possible PMS activation mechanism by FeCu-MOF and MB degradation mechanism.
- In the section introduction, the advances in photocatalysis should be improved, and the following articles are recommended to be cited
Appl. Surf. Sci. 486, 198–211; Appl. Catal. B Environ. 235, 225–237; J. Photochem. Photobiol. A Chem. 368, 168–181; Mater. Lett. 108, 72–74; J. Environ. Manage. 256, 109930.
Response: Thank the reviewer for the comments. As suggested by the reviewer, the relevant articles have been cited, please see our revised manuscript.
- Moderate English (language and style) improvement is required.
Response: Thanks very much for the comments. As suggested by the reviewer, we invited native English-speaking experts to checked the grammar and scientific writing of the manuscript.
- Please mention some possible disadvantages of your proposed method for decontamination of MB pollutants. (maybe in the conclusion part).
Response: Thank the reviewer for the comments. As suggested by the reviewer, the disadvantages of the method for decontamination of MB pollutants were added in the conclusion part, pleased see our revised manuscript in Page 16.
The possible mechanism based on both heterogeneous and homogeneous activation of PMS was proposed, as well as the MB oxidation mechanism. According to the MB intermediates, the catalytic performance needs to improve and mineralize the phenolic intermediates of MB to CO2 and H2O. From the viewpoint of engineering application, efforts are still needed to improving the water stability and reusability of FeCu-MOF in the future study.

Reviewer 3 Report
The manuscript titled " Synthesis of bimetallic FeCu-MOF and the performance as catalyst of peroxymonosulfate for degradation of methylene blue "
In this manuscript, authors reported that bimetallic MOFs (FeCu-MOF) was fabricated by hydrothermal process and used it to collect dangerous organic pollutants (MB) from an aqueous solution using photodegradation. The accomplished samples exhibited outstanding adsorption capabilities. Therefore, I would like to recommend published this work after addressing the following points:
1. Fig. 8, lack of error bar for the obtained data?
2. Introduction is well-organized and well-written, but the importance and novelty of the research should be highlighted and more clearly stated. The authors give some examples of works in the bibliography, but which is the advantage of their work in comparison with those works.
3. The authors are responsible for the English, which should be polished throughout the manuscript to clear some minor typo/grammar errors.
4. In the introduction part, Some publications are suggested to refer to improve the quality of the manuscript, such as: https://doi.org/10.1021/acsomega.1c03735, https://doi.org/10.1016/j.jtice.2021.08.034, https://doi.org/10.1007/s10904-022-02389-8.
5. All equation should be revised, which contain some typo error.
6. The author should better improve the beauty and quality of the figures in the manuscript.
7. The conclusion is also not targeted to the important aspects described in the manuscript; please rephrase it.
Author Response
Response to Reviewer #3
We very much appreciate the careful reading of our manuscript and valuable suggestions of the reviewer. Thank you for the kind advice. We have carefully considered the comments and have revised the manuscript accordingly. Below is our point-by-point response to the comments of Reviewer #2.
- Fig. 8, lack of error bar for the obtained data?
Response: Thanks for the Reviewer’s comment. As suggested by the reviewer, we have revised these errors. Thanks for pointing out. Please see our revised manuscript in Page 14.
Figure 9. (a) The variation of ferrous iron ions under different initial pHs; (b) The variation of total iron irons under different initial pHs.
- Introduction is well-organized and well-written, but the importance and novelty of the research should be highlighted and more clearly stated. The authors give some examples of works in the bibliography, but which is the advantage of their work in comparison with those works.
Response: Thanks very much for the comments. As suggested by this reviewer, we have revised the Introduction to highlight importance and novelty of our research, please see in Page 1, 2.
- The authors are responsible for the English, which should be polished throughout the manuscript to clear some minor typo/grammar errors.
Response: Thanks very much for the comments. We have checked many times and revised the typo/grammar errors in the text. In addition, we invited native English-speaking experts to checked the grammar and scientific writing of the manuscript.
- In the introduction part, Some publications are suggested to refer to improve the quality of the manuscript, such as: https://doi.org/10.1021/acsomega.1c03735, https://doi.org/10.1016/j.jtice.2021.08.034, https://doi.org/10.1007/s10904-022-02389-8..
Response: As suggested by the reviewer, the relevant articles have been cited, please see our revised manuscript.
- All equation should be revised, which contain some typo error.
Response: Thank the reviewer for the comments. As suggested by the reviewer, we have checked all equations many times and revised the typo errors in the text. Please see our revised manuscript.
- The author should better improve the beauty and quality of the figures in the manuscript.
Response: Thank the reviewer for the comments. As suggested by the reviewer, we have tried our best to improve the beauty and quality of the figures in the manuscript. Please see our revised manuscript.
- The conclusion is also not targeted to the important aspects described in the manuscript.
Response: Thanks very much for the comments. As suggested by the reviewer, we have revised the conclusion part to include important aspects described in the manuscript. Please see our revised manuscript.
This work demonstrated a strategy for synthesis of bimetallic MOFs catalysts for effective heterogeneous activation of PMS to remove organic contaminants from water. By introduction of copper ion into Fe-MOF, the obtained bimetallic FeCu-MOF exhibited remarkable efficiency compared with either Fe-MOF or Cu-MOF. 64 ppm MB (0.2 mM) can be 100% removed in the presence of 0.6 g/L FeCu-MOF and 6.0 mM PMS in 30 min. XRD, FT-IR, SEM, TEM, BET, and XPS were applied to characterize the physicochemical properties of the as-synthesized bimetallic MOFs which displayed the hydrangeas-like and rod-like structures with homogeneous distribution of unsaturated metal sites and an abundance of mesoporous. Effects of FeCu-MOF dosages, PMS concentration, initial MB concentration, initial pH, and common inorganic ions on the removal of MB were investigated. The results indicated that MB removal increased with the increase of the FeCu-MOF dosages and PMS concentration, but decreased with the increase of the initial MB concentration. Besides CHCOO−, other anions presented evident inhibitory effects on the removal of MB, which decreased in the order NO2− (0.0008 min-1) > H2PO4− (0.0114 min-1) > SO42− (0.014 min-1) > HPO42− (0.0152 min-1) > HCO3− (0.0173 min-1). The effect of initial pH on the MB removal ranked as follows: pH 3.16 > pH 9.05 > pH 5.33 > pH 6.96 > pH 10.87. Although SO4•−, ·OH, and 1O2 coexisted in FeCu-MOF/PMS system, SO4•− acted as the predominant reactive species for MB removal. The possible mechanism based on both heterogeneous and homogeneous activation of PMS was proposed, as well as the MB oxidation mechanism. According to the MB intermediates, the catalytic performance needs to improve and mineralize the phenolic intermediates of MB to CO2 and H2O. From the viewpoint of engineering application, efforts are still needed to improving the water stability and reusability of FeCu-MOF in the future study.

Round 2
Reviewer 3 Report
accepted in present form